# A Silicon-Compatible Synaptic Transistor Capable of Multiple Synaptic Weights toward Energy-Efficient Neuromorphic Systems

**Eunseon Yu [1], Seongjae Cho [2],\*** 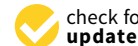 **and Byung-Gook Park [3],\***

[1]  Department of Electrical and Computer Engineering, Purdue University, West Lafayette, IN 47906, USA; yu966@purdue.edu

[2]  Department of Electronics Engineering, Gachon University, Seongnam-si, Gyeonggi-do 13120, Korea

[3]  Department of Electrical and Computer Engineering, Seoul National University, Seoul 08826, Korea

\*  Correspondence: felixcho@gachon.ac.kr (S.C.); bgpark@snu.ac.kr (B.-G.P.); Tel.: +031-750-8722 (S.C.); +02-880-7270 (B.-G.P.)

**Abstract:** In order to resolve the issue of tremendous energy consumption in conventional artificial intelligence, hardware-based neuromorphic system is being actively studied. Although various synaptic devices for the system have been proposed, they have shown limits in terms of endurance, reliability, energy efficiency, and Si processing compatibility. In this work, we design a synaptic transistor with short-term and long-term plasticity, high density, high reliability and energy efficiency, and Si processing compatibility. The synaptic characteristics of the device are closely examined and validated through technology computer-aided design (TCAD) device simulation. Consequently, full synaptic functions with high energy efficiency have been realized.

**Keywords:** energy consumption; hardware-based neuromorphic system; synaptic device; Si processing compatibility; TCAD device simulation

## 1. Introduction

Conventional computer architectures are mostly based on von Neumann's architecture since modern computer systems have been represented by electronic delay storage automatic calculator (EDSAC)—since 1949. The architecture consists of two main parts of processing and memory units performing the processes in the series' manner through single instruction and single data. Due to the physically differentiated system architecture, memory bus has been considered to be a bottleneck in determining the system processing speed, which is getting even worse in these days when big data are more increasingly demanded. In order to overcome this limit in the von Neumann computer architecture parallel processing capability of the artificial intelligent, of parallel processing with tremendous amount of data, contributions have been dedicated by the software-based neural networks. Although unimaginably many kinds of tasks have been accomplished by the software-driven technology in the given hardware system, with great resemblance to the way the human brain works, there is much room for enhancement of energy efficiency, which is the incomparable essence of biological system.

As a solution for the energy consumption issue, spiking neural network (SNN) is considered as one of the powerful schemes inspired by the biological system, which requires fundamental hardware innovation with synaptic transistors and neuron circuits [1,2]. Intellectual functions in human brain are determined by the strength and accuracy in connectivity among neurons. In human brain, there are a few tens of quadrillions of synapses and, through the synapses, humans become able to recognize, calculate, memorize, and learn. Thus, for hardware-driven neuromorphic systems to achieve more human-brain like computing efficiency, the synaptic device is required to have high

scalability, multi-level weight adjustability, large inference margin, strong tolerance, and ultra-low energy consumption. Moreover, in order to gain higher access to the chip-level production lowering time and cost barriers, the SNN should be realized on the Si platform being helped by the mature Si processing technology.

A number of synaptic devices have been proposed with memristors such as resistive-switching random-access memory (ReRAM) and phase-change random-access memory (PcRAM). They are considered to be good candidates for the electronic synapse owing to their high structure simplicity and volume scalability, mainly by their great geometrical resemblance to the two-terminal structure of the biological synapse and energy efficiency [3–6]. Although memristors have these advantages, there is still room for further improving the rather low endurance and reproducibility and for enhancing the completeness in realizing the biological synaptic functions. Moreover, some of the existing memristor devices are not in consideration of Si processing compatibility. The simple structure requires functional compensation by additional devices or circuits, which might cause increased overhead in the SNN architecture [7–9]. In this work, a novel synaptic device has been designed, which has SiGe quantum well (QW) and $Si_3N_4$ charge-trap layer to realize the short-term potentiation (STP) and long-term potentiation (LTP), respectively, and its synaptic operations have been validated through technology computer-aided design (TCAD) device simulation, Silvaco Atlas [10]. Although the designed synaptic device is in a more complicated structure with a larger number of terminals compared with the two-terminal synaptic devices, it is capable of complementing the aforementioned weak points of the memristors, with an emphasis on higher energy efficiency and Si processing compatible materials. While most of the memristor synaptic devices have shown energy consumption higher than 1 pJ [11], the largest energy consumption required for a potentiation event has been demonstrated to be 1.51 fJ.

## 2. Device Structure and Design Strategies

More detailed explanations on the operation principles of the synaptic device and the models used in the device simulation along with the related physics are provided as follows. Figure 1 shows the schematic of the proposed synaptic device which has a $p^+$ SiGe layer at the drain-side channel and a charge-trap layer on the channel.

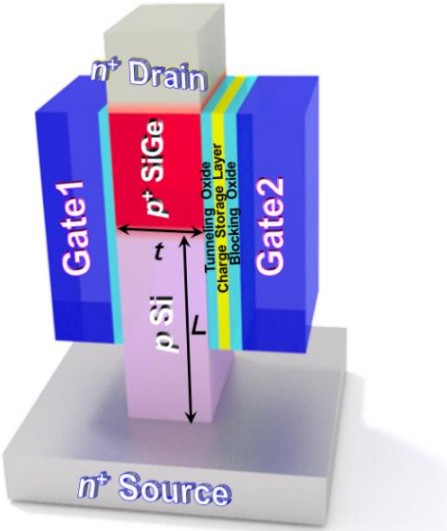

**Figure 1.** Schematic of the proposed synaptic device having an embedded SiGe quantum well and charge-storage layer for realizing the short-term and long-term plasticity, respectively.

As the number of potentiation pulses increases, the electrons in the SiGe valence band tunnel into the drain conduction band and fill the empty energy states. As a result, the holes generated in the SiGe layer are confined in the layer owing to a large valence-band offset (VBO) between Si and SiGe.

The confined holes give an effect of elevating the hole potential energy and the probability of hole tunneling into the nitride charge-trap layer, which realizes the LTP operation.

Designing a synaptic device with high reliability is paramount in building up a hardware architecture for the neuromorphic system. In order to demonstrate the device operation more accurately, multiple models are simultaneously activated. The mathematical and physical backgrounds of the used models can be glanced as follows. One of the essential differential equations used in the TCAD simulation is Poisson equation in Equation (1).

$$div(\varepsilon \nabla \psi) = -\rho \tag{1}$$

Here, $\varepsilon$ is the local electrical permittivity of the material, $\psi$ is the electrostatic potential, and $\rho$ is the volume charge density.

$$\frac{\partial n}{\partial t} - \frac{1}{q} div \vec{J}_n = G_n - R_n \tag{2}$$

Continuity equation in Equation (2) can be applied for obtaining the electron and hole current densities. $n$, $J_n$, $G_n$, and $R_n$ are concentration of mobile electrons, areal electron current density, generation rate of electron, and recombination rate of electron, respectively. $n$ can be substituted with $p$ for hole description. $q$ is the magnitude of electron charge. Based on the above equations, various models are equipped for higher accuracy and reliable simulation results. For an inversion layer mobility model, Lombardi model was used, which is suitable to non-planar devices, with dependences on both parallel and vertical electric fields, doping concentration, and temperature. The underlying physics comes from Matthiessen's rule.

$$\mu_T^{-1} = \mu_{AC}^{-1} + \mu_b^{-1} + \mu_{sr}^{-1} \tag{3}$$

Here, $\mu_T$, $\mu_{AC}$, $\mu_b$, and $\mu_{sr}$ indicate the total mobility, the surface mobility limited by scattering with acoustic phonons, the mobility limited by scattering with optical intervalley phonons, and the surface roughness factor, respectively.

$$f(E) = \frac{1}{1 + \exp\left(\frac{E}{kT_L}\right)} \tag{4}$$

For carrier statistics, Fermi–Dirac statistics was employed. In Equation (4), $f(E)$ is the probability that an available electron state with energy $E$ is occupied by an electron, $k$ is Boltzmann constant, and $T_L$ is lattice temperature. Moreover, the model is useful for the proposed device to describe the STP-to-LTP transition. The accumulated holes in the SiGe quantum well, which should be at the Fermi distribution tail, have higher probabilities of injection into the charge-trap layer. Moreover, non-local band-to-band tunneling calculation method was adopted, which has higher accuracy than the several tunneling models given as default in the TCAD simulation. This is due to the fact that the proposed device has the degenerately doped SiGe channel and drain, and the method calculates the tunneling probabilities by considering not only both forward and reverse tunneling currents but also the spatial variation of energy band and generation/recombination rates as shown in Equations (5) and (6).

$$J(E) = \frac{q}{\pi\hbar} \iint T(E)[f_l(E + E_T) - f_r(E + E_T)]\rho(E_T)dEdE_T \tag{5}$$

$$T(E) = \exp\left(-2\int_{x_{start}}^{x_{end}} k(x)dx\right) \tag{6}$$

Here, $J(E)$ is the net current density for a carrier with longitudinal ($E$) and transverse energy ($E_T$) under the assumption that the tunneling current is the result of bidirectional transfers of carriers across the junction. $f_l$ and $f_r$ are the Fermi–Dirac functions using the quasi-Fermi levels in the left-side and right-side materials of the respective junctions. $\rho(E_T)$ and $k(x)$ represent the density of states corresponding to the transverse wavevector components and the wavevector at $x$. $T(E)$ indicates the tunneling probability for a carrier having an energy of $E$ from the Wentzel–Kramers–Brilluoin

(WKB) approximation. Moreover, Shockley–Read–Hall recombination model, impact-ionization model, and bandgap narrowing model have been used. The aforementioned models are reflected for all the regions, and the non-local band-to-band model was applied locally between SiGe and Si where the tunneling events actually take place. In order to demonstrate the charge-trapping mechanism of nitride, a macro model (DYNASONOS) was employed, which includes various transport mechanisms such as thermionic emission, Poole–Frenkel emission, direct tunneling model, Fowler–Nordheim (FN) tunneling, and hot carrier injection at the same time. These models for the gate current are automatically applied for the $Si_3N_4$ layer and the regions in contact, which substantially affects the dynamics of the carriers moving into and out of the charge-trap layer. Without just using the default values given in the TCAD simulation package, the mobilities ($\mu$) [12,13], saturation velocities ($v_{sat}$) [14–19], bandgap energy ($E_g$) [20], and electron affinity ($\chi$) [21–27] of Si and SiGe have been fed into the device simulation [28]. This is because the SiGe layer, which stores holes, is considered as the important region for the synaptic operation. The values of the parameters are tabulated in Table 1.

**Table 1.** Parameters used in this work for Si and SiGe.

| | $\mu_n$ [cm/V·s] | $\mu_p$ [cm/V·s] | $v_{sat,n}$ [cm/s] | $v_{sat,p}$ [cm/s] | $\chi$ [eV] | $E_g$ [eV] |
|---|---|---|---|---|---|---|
| Si | 1590.0 | 570.00 | $1.02 \times 10^7$ | $7.33 \times 10^6$ | 4.050 | 1.10 |
| $Si_{0.7}Ge_{0.3}$ | 170.02 | 178.81 | $6.08 \times 10^6$ | $5.17 \times 10^6$ | 3.975 | 0.965 |

The SiGe layer is 50 nm long in the vertical direction and 50 nm wide (channel thickness = 50 nm). The *p*-type Si region is 100 nm long and the physical gate length ($L_g$) is 100 nm. Thus, whole SiGe region and the half of Si region are brought under the gate. In order to confine the holes generated over the potentiation process in the SiGe quantum well (QW) effectively, Ge fraction should be optimally controlled for a large valence-band offset (VBO) and the Si/SiGe interface status in the epitaxy processing as well, which is fixed to 0.3 throughout the design work. The gate oxide thickness for the gate 1 is 3 nm. The storage node is made up of oxide/nitride/oxide = 2/4/6 nm between the channel and the gate 2. The doping concentrations of source and drain junctions are both $n^+$-type $10^{20}$ cm$^{-3}$, and those of $p^+$ SiGe QW and *p*-type Si channel are $10^{18}$ cm$^{-3}$ and $10^{16}$ cm$^{-3}$, respectively.

## 3. Design Results and Discussion

### 3.1. Design of Synaptic Device

In designing the synaptic device, the focus was placed on successfully emulating biological neural system with Si compatibility, high scalability, high reliability, and high energy efficiency. In order to meet the requirements, various approaches were performed including embedding SiGe layer. There is a large difference in $E_g$ between Si and Ge and small difference in electron affinity ($\chi$) so that most of the difference in energy bandgaps is transferred to VBO, which forms a hole QW in the SiGe region. Furthermore, SiGe is not only helpful in implementing potentiation mechanism but also in large current ratio between different weight states because its smaller $E_g$ has the effect of lowering the potentiation voltage compared with the all-Si case. Employing these features of SiGe, the SiGe layer can be used as short-term storage node, making the device more energy-efficient.

Figure 2a shows the block diagram schematically explaining the learning rule of human brain by Hebbian's law [29]. Hebbian's law effectively dictates the correlation-based plasticity in the biological nervous system where the connectivity between pre-neuron and post-neuron, i.e., the synaptic conductance is strengthened by repeated firing events of the pre-neuron. An increased number of pulses in a given time, or equivalently, an increased pulse frequency enhances the transition probability of the synaptic device from short-term to long-term memory. Figure 2b,c shows the energy-band diagrams in the channel direction and metal-oxide-semiconductor direction from gate 1 to gate 2, respectively. For the potentiation operation, BTBT is adopted as the primary mechanism considering device reliability, scalability, and energy efficiency (Figure 2b). As shown in Figure 2b,

for a potentiation pulse, the valence-band electrons in the SiGe QW can see the empty states of the conduction band of the Si drain junction. As the result, holes are generated and effectively confined in the SiGe layer due to the large VBO between SiGe and Si. The locally confined holes by QW VBO give an effect of elevating the QW potential and increasing the channel conductance temporarily [30], which corresponds to the STP. Then, if the potentiation pulses are repeatedly applied to the transistor before the generated holes are annihilated by either recombination or diffusion, i.e., if the holes are accumulated and their amount exceeds a certain threshold value in the SiGe QW, LTP is introduced. The accumulated holes with the energies at the Fermi-Dirac distribution tail have higher probabilities of injection into the nitride charge-trap layer. Once the holes are trapped in the nitride layer, they do not vanish for long time, which establishes the LTP function. Moreover, work functions of those two gates are optimally adjusted to locate the BTBT site not in the vicinity of the right-side channel in order to prevent a soft potentiation and to store the generated holes at the right-side of the channel, which leads to a stable and reproducible LTP operation as shown in Figure 2c. By reflecting the aforementioned approaches, design of a synaptic device meeting the requirements is realized.

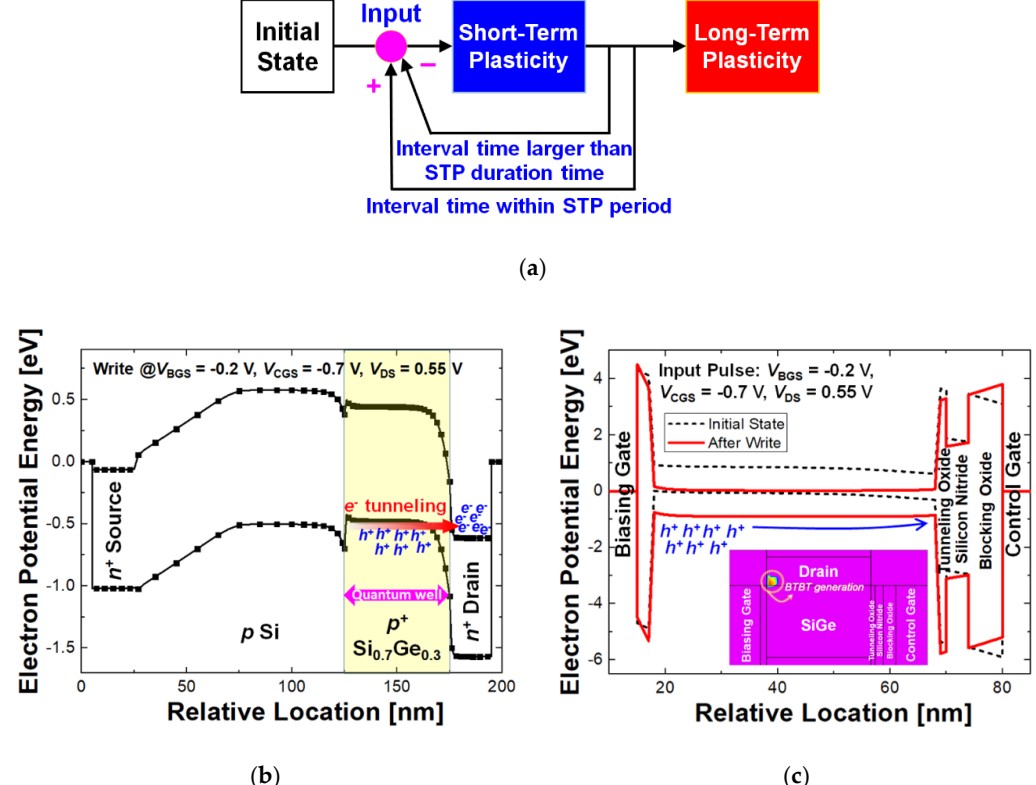

**Figure 2.** Operation principles of the synaptic device. (**a**) Hebbian's learning rule. (**b**) Energy-band diagram in the channel direction under the potentiation condition. (**c**) Energy-band diagram at the initial state and after potentiation state. The inset shows the band-to-band tunneling rate over a potentiation event.

### 3.2. Validation of Short- and Long-Term Plasticities

The proposed device has strong advantages particularly in energy-efficiency. There are many resources to make the device energy-efficient, such as introduction of SiGe QW, band-to-band tunneling mechanism, and STP characteristics. STP helps the device discriminate less important signals. Otherwise, when the weight of a synaptic device is changed at every input signal, the overall current over the synaptic device array would increase and large energy consumption is resulted.

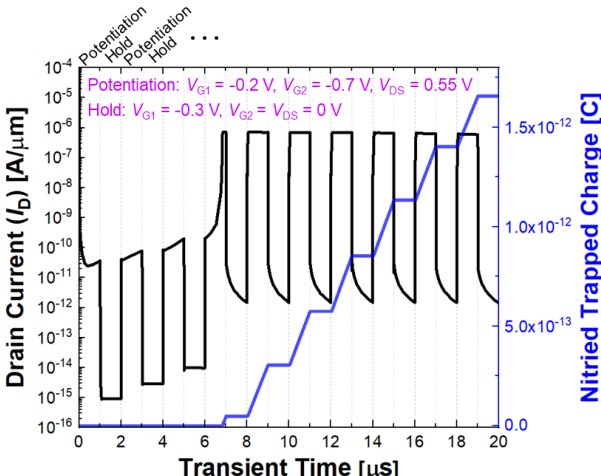

**Figure 3.** Drain current and nitride-trapped charges vs. learning pulses. The training pulse has a 1-μs width and a 1-μs interval.

Figure 3 indicates the timing diagram of drain current ($I_D$) and the amount of nitride-trapped charges as a function of time. The potentiation pulse is a set of (gate 1 voltage ($V_{G1}$), gate 2 voltage ($V_{G2}$), drain voltage ($V_{DS}$) = (−0.2 V, −0.7 V, 0.55 V), and the hold bias is ($V_{G1}$, $V_{G2}$, $V_{DS}$) = (−0.3 V, 0 V, 0 V). When a potentiation pulse is applied, holes are generated by band-to-band tunneling and confined in the SiGe layer. At the fourth pulse, $I_D$ rapidly increases since the number of holes in the SiGe exceeds a certain threshold value and induces a drastic injection into the nitride charge-trap layer as shown in Figure 3. The trapped holes lower the threshold voltage of the synaptic device and increase the channel conductance.

Figure 4a shows the conduction-band edges obtained after different number of pulses are applied: 0, 1, 5, 10, 20, 30, 40, and 50 pulses. The insets depict the three-dimensional (3-D) contours of conduction band edge surfaces at the initial state and at a state after 30 pulses are applied, respectively. The line spectra representing the conduction band edges have been extracted from the channel vicinity of $V_{G2}$ where the main current conduction path is formed. It is revealed that most of potential barrier lowering takes place by the holes in the SiGe region. Figure 4b plots the electron current density contours at the inference operations after different number of potentiation pulses: 1, 5, and 30 pulses. The inference process in the biological nervous system is analogous to the read operation in the memory array, and the electrical disturbance of the current data should be avoided. For the nondestructive inference, a voltage scheme was found to be $V_{GS1} = V_{DS} = −0.1$ V. As the number of pulses increases, more holes are populated in the charge-trap layer, and the potential barrier seen by the source electrons is lowered. Consequently, higher $I_D$ is read at the same inference voltage as can be confirmed by Figure 4b.

Figure 5a demonstrates the transient characteristics of the synaptic transistor after different number of potentiation pulses. Through Figure 5a, it is confirmed that the proposed synaptic device is capable of both STP and LTP functions. The STP increases the channel conductivity for a short time, and the effect is diminished as time passes. As a result, $I_D$ is eventually converged to the initial-state current level: The starting point can be varied but the final $I_D$ is the same in the STP operation. On the other hand, $I_D$ higher than the initial low current is consistently retained for up to $10^4$ sec or more

When the synaptic device is brought into the LTP states. Here, it is notable that a large current difference takes place between states as the number of potentiation pulses increases. In Figure 5b, the actual transfer curves of the synaptic device obtained after the corresponding different number of pulses are applied in Figure 5a are depicted. In the STP operation, there is steady-state threshold voltage ($V_{th}$) shift. Once the device is in the LTP condition, a larger number of pulses lead to lower $V_{th}$ without a temporal change. This is because the trapped holes in the nitride layer result in the inversion layer under the gate 2 at the inference bias. In Figure 5b, the proposed device demonstrates the large current ratio between high and low conductance states, which can be a beneficial aspect of a

fully-Si electron device. The successfully suppressed leakage current stems from the high potential barrier constructed by the large VBO. If there is only STP, there would be no $V_{th}$ shift. Only in the LTP condition, $V_{th}$ begins the left-shifts due to the holes trapped in the nitride layer. It is shown that $V_{th}$ of the proposed synaptic device is shifted by 1.5 V after 40 potentiation pulses are applied.

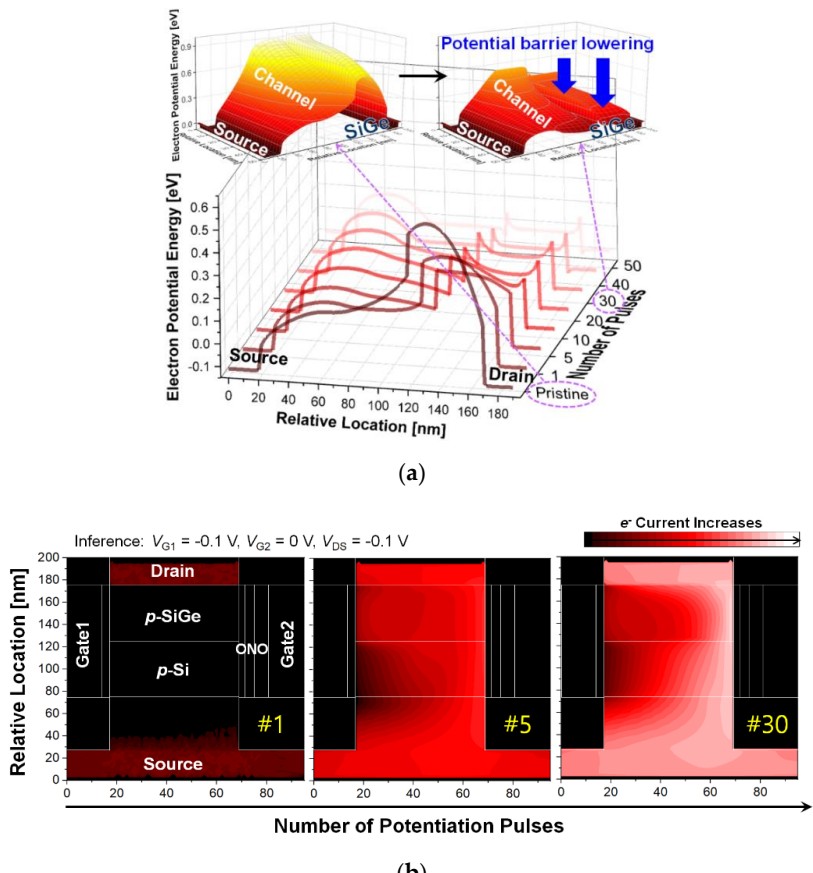

**Figure 4.** Analysis on potentiation operation. (**a**) Change in the conduction band surface with regard to the number of potentiation pulses: initial (left) and after 30 pulses (right). Line traces of the conduction band edges in the vicinity of gate 2. (**b**) Electron current densities after 1, 5, and 30 potentiation pulses applied to the synaptic device.

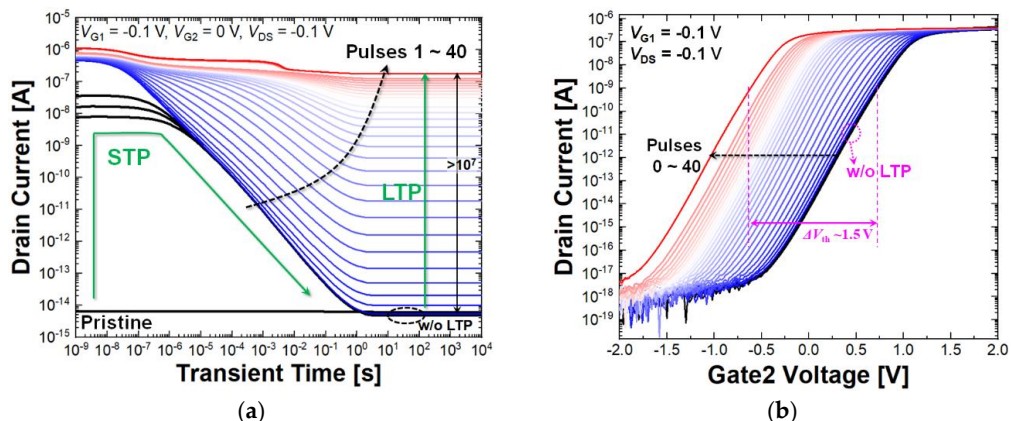

**Figure 5.** Electrical characteristics of the proposed synaptic device after different number of pulses: from 0 to 50 pulses. (**a**) Transient characteristics under read bias condition. (**b**) Transfer characteristics.

### 3.3. Interval Time Effects on STP and LTP Characteristics

Figure 6a–c shows how the transition from STP to LTP is made. As shown in Figure 6a, increasing the interval time between potentiation pulses makes it difficult to get into the LTP state. The holes in the SiGe layer temporarily generated by the pulses vanish by recombination and diffusion, which does not provide the boosting effect in band-to-band tunneling into the charge-trap layer. With the interval time of 1 ms, the synaptic device is not allowed to move to the LTP states as shown in Figure 6a and confirmed by Figure 6b. Figure 6b demonstrates the transient and DC characteristics under different interval time conditions for the same total number of potentiation pulses of 10. It is assured that a short enough time interval allows the synaptic device to enter the LTP states and modulate the electrical conductivity for learning.

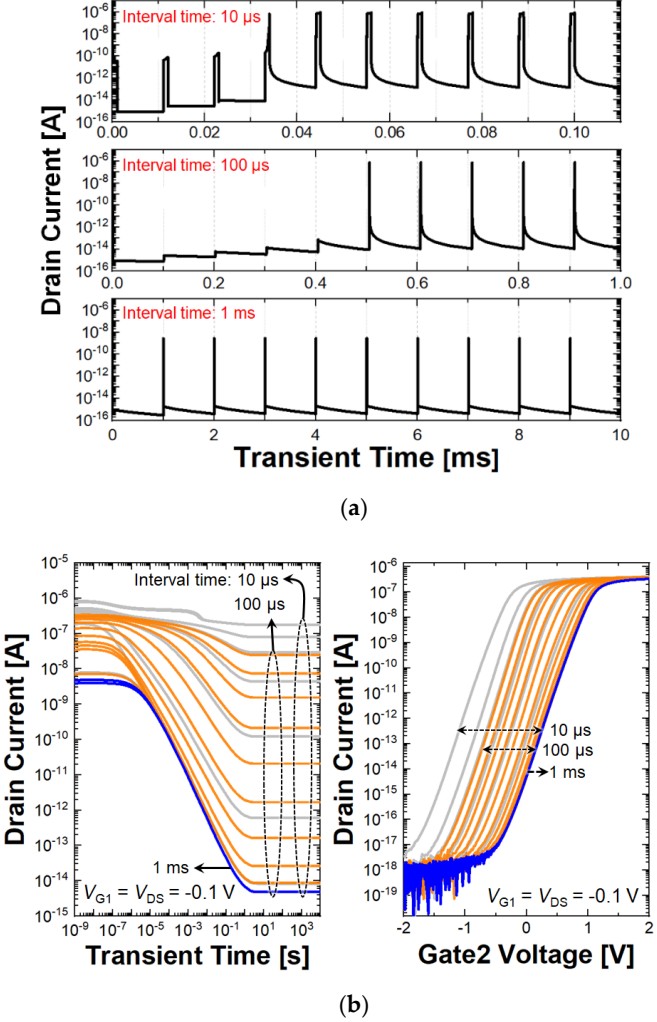

**Figure 6.** Operation characteristics depending on the number of pulse interval times. (**a**) Current changes with different pulse interval times: 10 µs, 100 µs, and 1 ms. (**b**) Transient (left) and DC sweep (right) characteristics with different interval times. For a shorter interval time at a given number of potentiation pulses, the saturation current increases and the $V_{th}$ shift gets wider.

### 3.4. Spike-Timing-Difference Plasticity and Array Architecture

Table 2 summarizes the bias conditions for the synaptic operations and the required energy consumption per the realizable synaptic event along with the calculated synaptic device density. In order to exactly calculate the energy consumption, the current is integrated with time and multiplied by voltage. In Table 2, the energy consumption was obtained in the case of maximum value to consider

the worst case. There are approximately $10^{15}$ synapses in the brain, about 1% of them are activated at the same time, and the frequency of neuron spike is about 10 Hz [31–33]. On this bases, human brain consumes power of ~20 W, and 55% out of the total power is consumed by the action potential [34–37]. In other words, it is assumed that power of 11 W is consumed for the synaptic activities. The power consumption per biological synaptic event is derived to be $1.1 \times 10^{-12}$ W. Considering that each synaptic event has a ~100 ms duration, the energy consumption per synaptic event is calculated to be about 10 fJ. For the inference operation, 20 ns of rising and falling times and 10 ns of pulse duration are schemed, and then, this time period is multiplied by the inference voltage of −0.1 V and the current depending on weight for calculation of the energy consumption. The energies required for respective synaptic operations are summarized in Table 2, and most of them are very close to those for the biological synapse. Potentiation operation increases the conductance of the synaptic device so that a relatively large energy consumption is required; however, the amount is still low. Owing to the low operation voltage and tunneling-based injection mechanism, maximum energy consumption of only 0.52 pJ is needed for a potentiation event. For a depression event, the trapped holes tunnel back to the channel, which necessitates a relatively high operation voltage on the gate 2. However, even in the worst case, the required energy consumption is much lower than that for a potentiation event. Although the energy consumption for individual potentiation or depression event can be higher than that for an inference operation, the energy for an inference event can be spanned over a large range depending on the conductance. With the help of low-power and high-speed operation capabilities, the proposed synaptic transistor requires only a femto-joule energy even after 40 potentiation pulses are applied. Assuming that a unit cell has a footprint of 158 nm by 150 nm, the density of synaptic device array is calculated to be $9.09 \times 10^9/cm^2$. Here, the critical dimensions of the designed device and the metal pitches in one of the most recent memory technologies have been considered [38], where the wordline (WL) and bitline (BL) pitches are 48 and 54 nm, respectively.

**Table 2.** Energy consumption per realizable synaptic event and the calculated synapse density.

| | $V_{G1}$ | $V_{G2}$ | $V_{DS}$ | Time | Energy | |
|---|---|---|---|---|---|---|
| Potentiation | −0.2 V | −0.7 V | 0.55 V | 1 μs | | 0.52 fJ |
| Depression | 0 V | 6 V | 0 V | 1 μs | | 1.51 fJ |
| Inference | −0.1 V | 0 V | −0.1 V | 10 ns | Initial | $6.42 \times 10^{-24}$ J |
| | | | | | 20 pulsed | $1.87 \times 10^{-16}$ J |
| | | | | | 40 pulsed | $5.24 \times 10^{-16}$ J |
| Synapse density | $9.09 \times 10^9/cm^2$ (∵Unit cell size: 95 nm × 117 nm) | | | | | |

Furthermore, the spike-timing-dependent plasticity (STDP) characteristics have been obtained by adjusting the pulse profile and time difference as demonstrated in Figure 7. The inset describes the potentiation and depression pulse schemes in the STDP simulations. The spiking pulse has 950 ns of rising time and 100 ns falling time, respectively, and the positive and negative voltage peak is 0.72 V in magnitude. The pre-neuron is connected with drain, and the post-neuron is with gate 1 and gate 2. When a pre-neuron spike comes earlier than a post-neuron one, the synaptic transistor is potentiated since the holes generated by tunneling operation from channel to drain are stored in the nitride charge-trap layer, which improves the conductance of the device. In the reversed order, the device is depressed owing to ejection of the trapped holes out of the nitride layer. When the timing difference between pre- and post-neuron spikes is larger than 900 ns, the conductance of the device is not changed but left as the initial value, which indicates that two neurons are not so closely correlated, and there is neither potentiation nor depression. It is noticeable that the synaptic transistor has a large current difference for the different spike timing, which substantially reduces the complexity of the sensing circuits and enhances the system reliability. Figure 8 demonstrates a presumable array architecture with the designed synaptic transistor for a hardware-driven neuromorphic system.

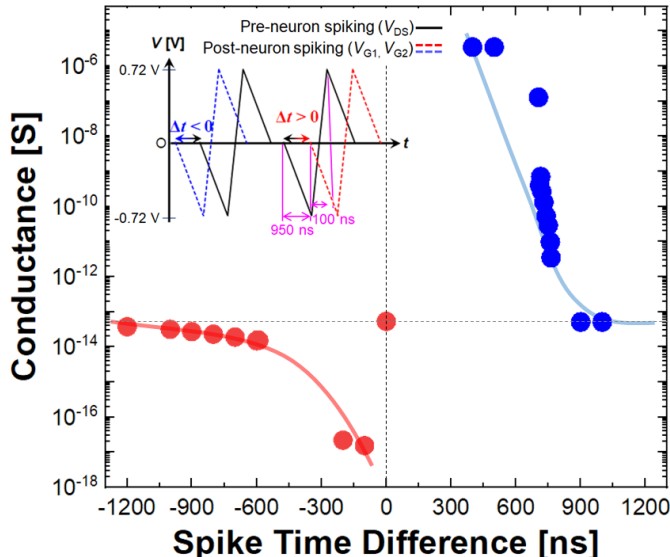

**Figure 7.** Spike-timing-dependent plasticity characteristic of the proposed synaptic device. The inset describes the potentiation and depression pulse schemes.

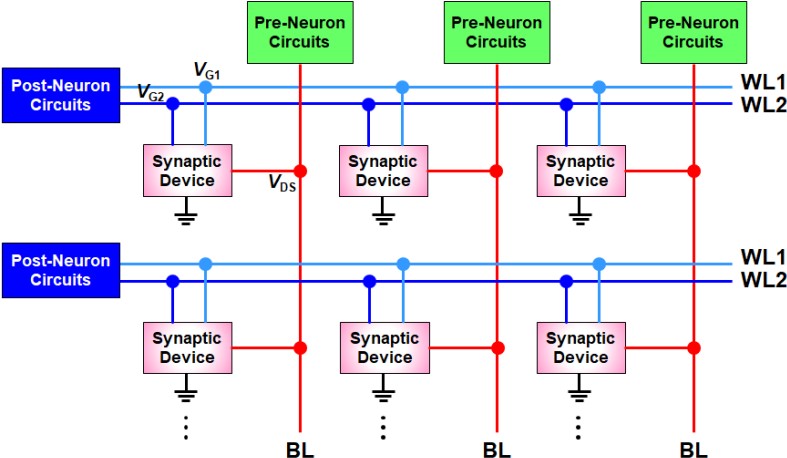

**Figure 8.** Array architecture with the proposed four-terminal synaptic transistor towards a high-density and high-reliability hardware-driven neuromorphic system.

## 4. Conclusions

In this work, a synaptic transistor having SiGe quantum well and nitride charge-trap layer was schemed and characterized by a series of rigorous simulation works. The synaptic device has successfully demonstrated the synaptic operations including STP, LTP, and inference with high energy efficiency not exceeding a two femto-joules in the worst case. Further, spike-timing-dependent plasticity was verified through a properly adjusted pulse scheme. A presumable array architecture is also conceived with the four-terminal synaptic device, and its density was calculated to be $9.09 \times 10^9/\text{cm}^2$ based on the interconnect schemes in the 18-nm DRAM technology node.

**Author Contributions:** E.Y. and S.C. conceived the device structure and wrote the manuscript. E.Y. performed the device simulations and evaluated the device scalability and array density. S.C. approved the simulation results in confirmation of the biological analogies. B.-G.P. conceived and developed the various types of hardware-driven neuromorphic systems, initiated the overall research project, and confirmed the validities of the simulated synaptic operations towards the artificial spike neural network.

**Funding:** This work was supported by Nano·Material Technology Development Program through the National Research Foundation of Korea (NRF) funded by the Ministry of Science and ICT (MSIT) (Grant

No. NRF-2016M3A7B4910348) and by Mid-Career Researcher Program through NRF funded by the MSIT (Grant No. NRF-2017R1A2B2011570).

**Conflicts of Interest:** The authors declare no conflict of interest.

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
