# Peer review of "A Silicon-Compatible Synaptic Transistor Capable of Multiple Synaptic Weights toward Energy-Efficient Neuromorphic Systems"

_electronics, doi:10.3390/electronics8101102_

Round 1

Reviewer 1 Report

Authors believed that they designed a reliable synaptic device with high performance as hardware architecture for the neuromorphic applications. In overall terms the manuscript is well-explained, however the manuscript can benefit from substantially improvements before it is accepted for publication:

1. Authors are recommended to update the introduction with cons/pros of floating gate transistor synapse device compared with the memristor ones.

2. Figure 1a represents synaptic Hebbian rules but neither in caption nor in the manuscript the diagram is defined.

3. Figure 7 is one of the most important figures of the manuscript with the least explanation.

4. Please explain what is the inference operation mode and corresponding energy calculation which give similar values as biological synapses.

5. Is there any validation/comparison of the current results with a real physical device?

Authors believed that they designed a reliable synaptic device with high performance as hardware architecture for the neuromorphic applications. In overall terms the manuscript is well-explained, however the manuscript can benefit from substantially improvements before it is accepted for publication:

1. Authors are recommended to update the introduction with cons/pros of floating gate transistor synapse device compared with the memristor ones.

2. Figure 1a represents synaptic Hebbian rules but neither in caption nor in the manuscript the diagram is defined.

3. Figure 7 is one of the most important figures of the manuscript with the least explanation.

4. Please explain what is the inference operation mode and corresponding energy calculation which give similar values as biological synapses.

5. Is there any validation/comparison of the current results with a real physical device?

Author Response

Dear the Reviewer, please find the attached answer sheet for your questions and comments. Thank you.

Reviewer 2 Report

The authors present an interesting device for a synaptic transistor. The paper need to be improved in many areas.

The explanation of the TCAD models and physics used is not described well enough. The one technical paper referred to for the TCAD does not employ the same structure.

The structure seems very unique. The authors need to describe the feasibility of processing the structure.

There are not enough references in this paper.

It would be useful to the readers to compare the proposed device to others used in neuromorphic computing. The introduction needs to be augmented.

There are many articles (the, a, an, etc.) missing in the sentences. Please check for correct English language style.

Author Response

Dear the Reviewer, please find the attached answer sheet. Thank you.

Round 2

Reviewer 2 Report

I thank the authors for the edits on the first submission. I suggest more edits to make this paper useful and well-accepted by the community.

line 37: "for enhance the energy efficiency" show be "to enhance" or "for enhancement of"

line 43: this wording "successful implementation of the human brain" implying something different than what was intended. I believe you want to say something like " for hardware-driven neuromorphic systems to achieve human brain-like computing efficiency, ... The way it is written sounds like you want to make the human brain not just mimick it.

line 48: the new sentences added should be in a separate paragraph.

lines 52-54: the sentence in these lines is a run-on sentence. Please rephrase for correct grammar.

Many articles are still missing throughout.

Since this work is fully based on simulation results, the TCAD methods need to be much better explained. The edits were not enough.

For example, as a researcher with TCAD expertise, I want to see the name of the simulation tool written out in the text (Silvaco Atlas) in addition to the reference and why that tool was chosen.

Also, it is common in TCAD to specifically state the differential equations used to describe the physics used in the simulation and not just the parameter models (i.e. band-to-band tunneling). The physics that Silvaco uses is the Poisson equation for electrostatics and continuity equations for electrons and holes, I believe. Check the user manual.

The authors also need to give reasons why "

including concentration and field-dependent mobility models, Fermi-Dirac carrier statistics model, Shockley-Read-Hall recombination model, bandgap narrowing model, impact ionization model, and non-local band-to-band tunneling calculation"

are needed for more accurate models. Sometimes they are, sometimes they are over-kill.

Please state the values used for these parameters (

mobility, saturation velocity, bandgap energy, and electron affinity of SiGe empirically obtained as a function of Ge fraction have been fed into the device)

with seminal references for each.

Also in the methods section, it would be nice to give an overview of how the device works and where exactly the physics were implemented. For example, band-to-band tunneling is the dominant mechanism of the potentiation. (I would also like to see the physical equation that Silvaco uses). Is the model implemented on both gate 1 and gate 2? It is unclear.

Currently the methods section is less than one page. I would suggest to expand it to two or more pages so that you convince the readers that you are implementing the physics properly.

Author Response

Please find the attached answer sheet to the reviewer. Thank you.

Round 3

Reviewer 2 Report

I thank the authors for the improvement of the TCAD methods section. It makes the paper much more robust.

I have highlighted one sentence that needs to be rewritten for clarity.

In order to help understanding the simulation approached, explanations on the device

operations more are given here more in detail.

Author Response

We the authors thank the reviewer for the kind comment. According to the Reviewer's comment, we have corrected the original sentence for higher clarity in expression.

From line 72 to 73 on page 2:

More detailed explanations on the operation principles of the synaptic device and the models used in the device simulation along the related physics can be provided as follows.

This is the introduction line for the Chapter 2 in which most of the contents are dedicated to explain the operation principles and simulation models with their mathematical and physical significance.

Thank you so much for your valuable time and comments generously provided for reviewing our manuscript.

- Sincerely, Seongjae Cho, the Corresponding author.